# Analysis of the Water Leakage Rate from the Cells of Nursery Containers

**Mariusz Kormanek** [1,*] and **Stanisław Małek** [2]

[1] Department of Forest Utilization, Engineering and Forest Techniques, Faculty of Forestry, University of Agriculture in Krakow, Al. 29 Listopada 46, 31-425 Kraków, Poland

[2] Department of Ecology and Silviculture, Faculty of Forestry, University of Agriculture in Krakow, Al. 29 Listopada 46, 31-425 Kraków, Poland; stanislaw.malek@urk.edu.pl

[*] Correspondence: mariusz.kormanek@urk.edu.pl

**Abstract:** In container production, the key issue is proper irrigation and fertilization. Typically, the water required for plant growth is supplied through an irrigation ramp system, which can also perform fertilization. The frequency of irrigation and the amount of water supplied by the ramp depends on several factors, such as the species of plants grown, the container used, the substrate, and atmospheric factors accompanying production. For effective irrigation, the substrate in the container cell must retain the supplied water long enough for plant absorption. However, any excess water should drain from the container. To optimize irrigation, it is important to determine the parameter of the water outflow speed from the container cell, which is difficult to determine. This work proposes a new solution for a station that can measure the water outflow speed from various container cells (patent application P.443675 2022). In tests, the water outflow speed was assessed for two Styrofoam container types (V150—650/312/150 mm, 74 cells, and 0.145 dm$^3$ cell volume; and V300—650/312/180 mm, 53 cells, and 0.275 dm$^3$ cell volume). Both were filled with a peat and perlite substrate (95/5%) using the Urbinati Ypsilon line (V150 substrate moisture 75.7 ± 1.1%, and V300 75.9 ± 2.1%, efficiency of the line 400 containers·h$^{-1}$, vibration intensity of the vibrating table—maximum acceleration 12 G). The results indicated that the water outflow speed varied between container types. The V300 container had a higher outflow speed (0.0344 cm·s$^{-1}$) compared to the V150 (0.0252 cm·s$^{-1}$). This discrepancy may be due to differences in dry bulk density, with a correlation of $r = -0.523$. The V300 had a lower actual and dry bulk density (0.418 g·cm$^{-3}$; 0.079 g·cm$^{-3}$) compared to V150 (0.322 g·cm$^{-3}$; 0.103 g·cm$^{-3}$). This highlights the need for individual selection of parameters on the backfilling line for different container types when filling. Using identical parameters for diverse containers can lead to varying substrate volume densities, impacting water outflow rates.

**Keywords:** substrate; container; Darcy's law; strain gauge; seedling

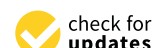



## 1. Introduction

Growing seedlings in containers is a specific method of plant breeding, separate from traditional agricultural production. For agricultural crops, water supply is based on natural rainfall, and annual fertilization is based on the expected yield, as well as on the characteristics of the soil, which determines the use of fertilizer for various plant species [1]. In contrast, container production uses artificial irrigation, often through ramp systems, and incorporates fertilizer directly into the substrate or irrigation water. The production system depends on various factors. Besides the species being grown and the substrate used, other elements like the density of seedlings in the container, container size, method of fertilizer delivery (whether the fertilizer is dissolved in the substrate or on the surface), and the date of fertilizer application are crucial. One issue is that fertilizer dissolved in the substrate can easily wash out during intensive irrigation [1–3]. Irrigating

containers using a ramp involves cyclical movement over the containers and the time during which a large mass of water is delivered to the container surface in a relatively short span. This results in the substrate in the containers experiencing repeated wetting (from irrigation) and drying (from evaporation and root water uptake). The water and air conditions in the growth medium, thus, fluctuate throughout the day and across the growing season. These frequent and sudden changes affect water and oxygen retention and water outflow from the container. The availability of water and air is determined by the retention characteristics of the substrate, as well as by the environmental conditions during seedling production [2,4]. In Poland, the preferred substrate for seedling production mainly comes from high sphagnum peat, sometimes mixed with components like perlite and vermiculite. Peat offers high porosity, water capacity, and sterility, and has a low mineral content, simplifying fertilizer dosing [3]. Using only peat can sometimes lead to a high level of water retention and shrinkage upon drying. As per Heiskanen's research [2], this might result in overflow and insufficient air, if watering reaches the container's water capacity. The parameters for the commonly used peat–perlite substrate in Poland are total porosity 70%–93% by volume, water capacity at 73% by volume, air between 20% and 25% by volume, available water at 48% by volume, with a wet weight of 864 $kg \cdot m^{-3}$ [3,5–7]. These ranges are extensive, and the measurement methods can be ambiguous, complicating real-time control during seedling growth [8–11]. If the substrate's physical parameters are inappropriate, modifying them becomes challenging. The main problem is too-low or too-high air capacity, typically linked to the substrate's density [11,12]. The literature indicates that degree of substrate compaction and settlement might increase with the use of larger substrate elements, varied bulk densities of component, and intense irrigation. The substrate's density changes over time and is the result of the movement of small substrate particles from the upper level of the cell to the lower one and the decomposition of organic matter. This is accompanied by a decrease in porosity, air, and water available to plants. Root growth can simultaneously increase substrate density but also boost its permeability, facilitating gas diffusion [4,7,11–16]. The need to provide water quickly to plants, especially when they are growing in a limited amount of substrate within a confined container space, requires that water does not flow out from the container and that the plant has time to absorb it. In this case, the speed of liquid outflow from the container cells is crucial for plants to absorb both water and water with fertilizers. If this speed is too high, the plants will not have time to use the supplied water and fertilizers, which will result in the need for additional irrigation. This can reduce the efficiency of the irrigation and fertilization process, increase production costs, and pose environmental risks since runoff with fertilizers can enter into groundwater [1]. The drainage speed is influenced by the container's type, substrate, and filling method—usually performed on automated lines with tools like vibrating tables, scraper brushes, or pressure fingers. The aim of this work was to determine the influence of the container type and filling parameters on the speed of liquid outflow from the container. For this, a new prototype measuring station was utilized [17]. The device allows simultaneous measurement in several container cells.

The aim of this research was to determine the speed of water outflow from substrates within containers of varying cell sizes, as well as the variability of this parameter within individual containers. The following research hypotheses were considered: (1) the speed of water outflow from the substrate in a container remains consistent, irrespective of the container cell's size, provided the container filling conditions are identical; (2) no differences exist in water drainage through the substrate within a single container.

## 2. Materials and Methods

### 2.1. Preparing Material for Testing

The containers were filled with substrate on 21 April 2022, on the automatic line of Urbinati S.r.l. Ypsilon (Figure 1) at the Nursery Farm in Suków Papiernia (50.79613, 20.71011), Daleszyce Forest District. The experiment utilized V150 and V300 Styrofoam containers manufactured by Marbet (Table 1, Figure 2).

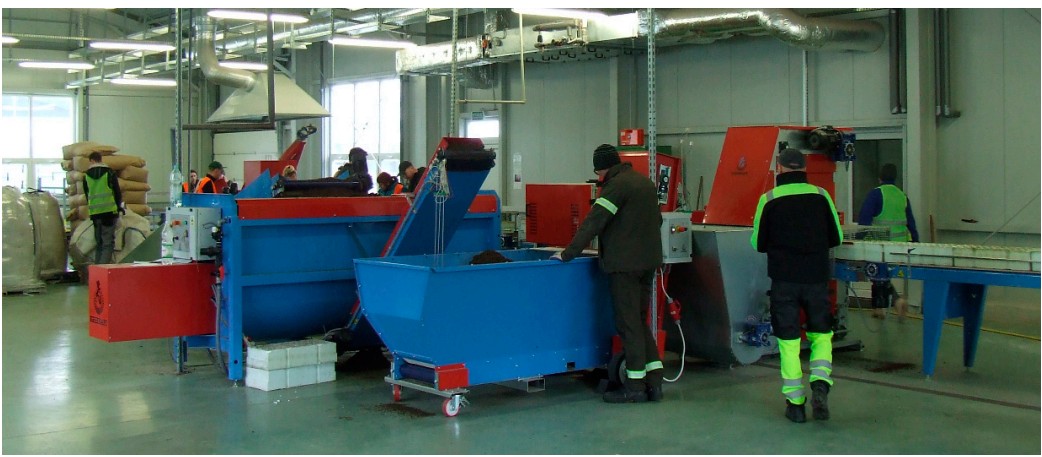

**Figure 1.** Urbinati S.r.l. automated line at the Nursery Farm in Suków. Photo: M. Kormanek.

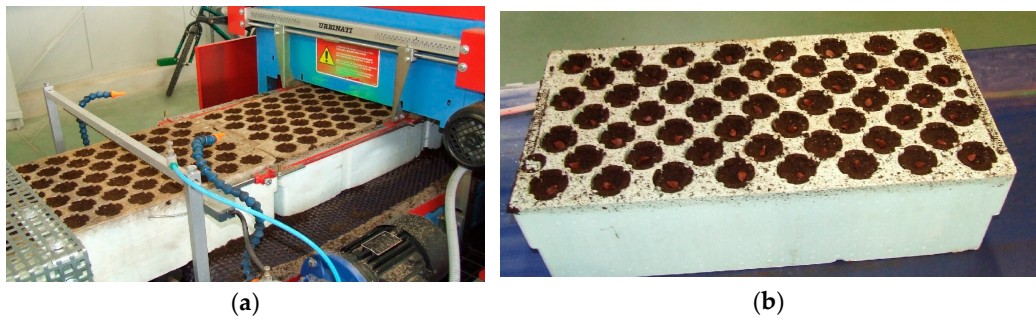

(**a**)                     (**b**)

**Figure 2.** Filling the V150 container with substrate (**a**) and the V300 container filled with substrate and sown (**b**). Photo: M. Kormanek.

**Table 1.** Parameters of the containers used in the experiment.

| Parameter | Container Type | |
|---|---|---|
| | **V150** | **V300** |
| Length/Width/Height $L/W/H$ | 650/312/150 mm | 650/312/180 mm |
| Number of cells $nc$ | 74 pc. | 53 pc. |
| Cell volume $V$ | 0.145 dm$^3$ | 0.275 dm$^3$ |
| Cell height $H$ | 15.0 cm | 18.0 cm |
| Diameter of the entrance hole to the cell $Din$ | 4.6 cm | 5.2 cm |
| Diameter of the outlet hole from the cell $Dout$ | 2.5 cm | 2.5 cm |
| Average flow area $A$ | 10.4 cm$^2$ | 13.1 cm$^2$ |

These type of containers are commonly used in Poland for the production of coniferous species, e.g., pine, spruce (V150), and deciduous species, e.g., beech and oak (V300).

In these containers, the cells with the same diameter as the entrance hole to the cell—*Din* have a shape similar to a truncated cone, and root guides are made along the side walls of the cell. In the lower part of each cell there is a narrowing that prevents the substrate from spilling out with an outlet hole with a diameter *Dout*. The containers were filled with peat–perlite substrate (95/5 by volume) that had the following granulometric composition: fraction 10.1–20 mm: 2.5%, 4.1–10 mm: 12.5%, 2.1–4.0 mm: 12.5%, <2.0 mm: 72.5%. The substrate's maximum degree of decomposition was 15%, and its organic matter content was >85%. This substrate is produced in Poland based on imported peat, and the percentage of perlite added is determined individually for each batch of peat delivery, based on the analysis of air and water capacity.

Before filling the containers, the substrate was moistened in a line mixer equipped with spray nozzles. Moisture was organoleptically controlled during the moistening process by the line staff. They adjusted the moisture of the substrate to the level typically used when filling containers. For the experiment, after moistening and while refilling the line's buffer tank, 12 substrate samples were taken (comprising four series of three samples each). Their moisture content was determined in % on a WPS 110 dryer scale, accurate to ±0.1%. The moisture content of the substrate when filling V150 containers was 75.7 ± 1.1%, and for V300, it was 75.9 ± 2.1%. The vibration intensity of the vibrating table remained constant during the filling, regardless of the container type. This amounted to 12.0 G of maximum acceleration, measured using the Voltcraft DL-131G device, with an accuracy of ±0.5 G [18]. Throughout the tests, the line's efficiency remained consistent, set at $E = 400$ (containers·h$^{-1}$), which is the typical rate at which containers are filled at this nursery. The operating parameters of the line, containers and substrata, were the same as those used in the experiment performed earlier [19].

### 2.2. A Prototype Test Stand for Testing the Outflow of Liquids through Container Cells

To measure the outflow of liquid from the substrate in the container cells, a prototype measurement station was used (Figure 3), registered with the Patent Office of the Republic of Poland under the number P.441918 [18]. This is an automated measuring device that can measure the outflow of liquid from the substrate in multiple container cells simultaneously. The station comprises a main frame (1) in which the lower (2) and upper (3) tanks are installed. A horizontal fixed frame (4) is mounted in the central part of the main frame, and a replaceable drain plate (5) is inserted into it. The test container (6) is placed atop this drain plate. The plate features drainage connections, which allow liquid to drain from the bottom of the individual cells of the container (6). The spacing of the drain stubs is consistent with the spacing of cells in a given type of container. From the top, the container is pressed by a movable frame (7) mounted on linear bearings (8) that traverse vertical guide rollers (9). The frame is pressed by pressure springs (10), with the tension caused by the lever mechanism (11). To ensure there is no liquid leakage or wastage, seals are positioned between the upper frame and the upper surface of the container and between the lower surface of the container and the drain plate. Water is supplied to the upper frame from the upper tank (3) via adjustable spray nozzles (12). An overflow hole in the upper frame ensures a steady water level over the upper surface of the container. Any excess liquid flows into the lower tank (2). Water is pumped from the lower tank (2) to the upper tank (3) using a pump (13), ensuring the upper tank (3) and movable frame (7) maintain a full water level, essential for consistent measurement conditions. From the drainage outlets on the drain plate (5), the liquid moves through settling filters (14) and is directed by solenoid valves (15) and (16) either to the lower tank (2) or to measuring containers (17) that sit atop strain gauge weighing sensors (18). Settling filters (14) consisted of a small cylindrical tank with an inlet pipe inserted from the top and outlet pipe at the side of the tank. During flow, pollutants accumulate at the bottom of the tank. Data about the water's weight, collected from the strain gauge sensors (18), are relayed to the recording system that comes with the station. In this setup, during a measurement, the drain outlets individually collect liquid from 45 cells. The liquid from three such cells is funneled via a four-way device to 1 of 15 measuring containers placed on the weighing sensors.

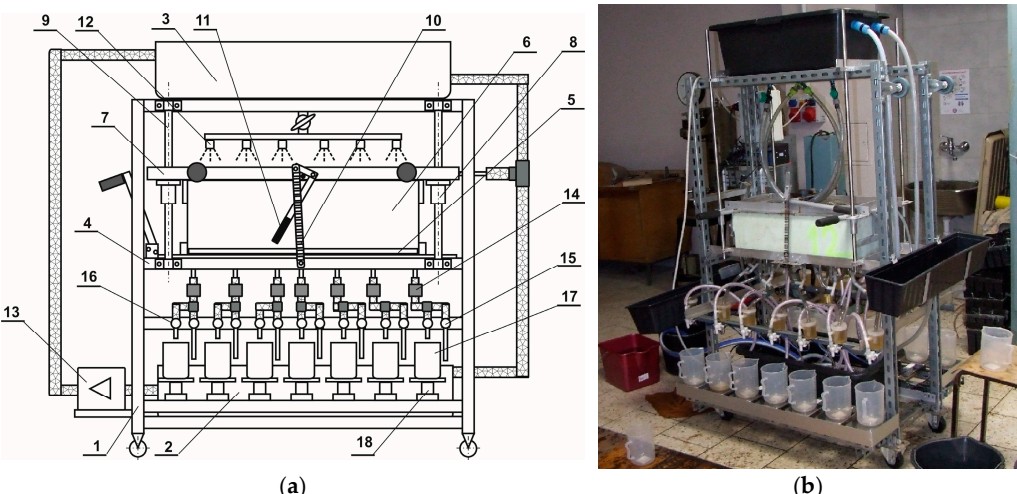

(a)　　　　　　　　　　　　　　　　　　　　(b)

**Figure 3.** Prototype station for measuring the rate of water leakage from a nursery container [18], diagram of the station (**a**); measurement in the container (**b**). Photo: M. Kormanek 1—main frame, 2—lower tank, 3—upper tank, 4—fixed frame, 5—drain plate, 6—container, 7—movable frame, 8—linear bearings, 9—guide rollers, 10—compression springs, 11—lever mechanism, 12—spray nozzles, 13—pump, 14—settling filters, 15, 16—solenoid valves, 17—measuring containers, 18—strain gauge sensors. Figure and photo: M. Kormanek.

### 2.3. Measurement Process

During the testing material preparation, 10 V150 containers were initially filled to stabilize the line's operating conditions. Subsequently, another set of 10 V150 containers were filled. From this set, the first four containers were used for bulk density (*BD*) measurements, the next two were designated for outflow measurements on the prototype station, and the last four were used for density measurements (*BD*). When the line was transitioned to filling V300 containers, 10 containers were filled to stabilize the operating conditions. The subsequent 10 V300 containers measured similarly to the V150. For volumetric density measurements, containers had a collector placed on their upper surface. This collector had either six holes for V150 or five for V300 containers. These holes were evenly distributed on the surface of the container (positioned diagonally and in the middle of the longer sides of the containers). Volume cylinders, with a 500 mL capacity, were inserted into these holes (Figure 4a). Subsequently, the container, with the collector, was rotated 180 degrees. This action caused the contents of selected cells (substrata) to spill out into the cylinders (Figure 4b). The cylinders into which the substrate from a single cell fell were weighed to determine the mass of the wet substrate in individual cell *mw* [g]. Knowing the volume of a single cell *V* (cm$^3$) (Table 1) and the mass of the substrate *mw*, the actual *ABD* (g·cm$^{-3}$) of the substrate in selected cells of the V150 and V300 containers was calculated. Then, the substrate collected from individual cylinders was dried at 65 °C for 48 h to obtain the dry mass *md* (g), and taking into account the cell volume *V* (cm$^3$), the dry bulk density *DBD* (g·cm$^{-3}$) was determined.

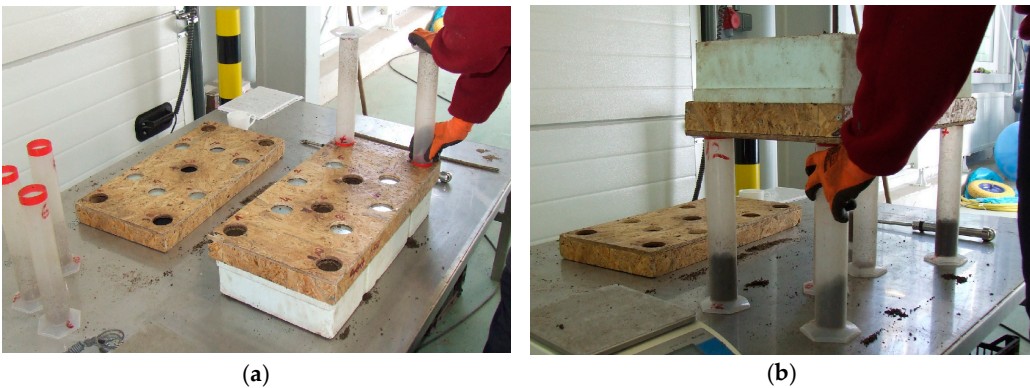

**Figure 4.** Measurement of bulk density in container cells. A collector with volumetric cylinders mounted on a container (**a**), pouring the substrate into the cylinders (**b**). Photo: M. Kormanek.

The water outflow measurement from individual cells of the V150 and V300 containers involved several steps. Initially, a single container was mounted on the measuring station. Water was then supplied from the upper tank through a spray nozzle to the upper movable frame. This frame pressed against the container from above, facilitating the soaking of the substrate in the cells. Once the substrate was soaked, stabilization of the outflow was achieved by letting the liquid flow freely through electro-valves into the lower tank. This process lasted for 1 h. Following this, water was supplied to the container for the next 1 h. Afterward, the liquid drained from the drain stubs using solenoid valves into tanks situated on strain gauge load cells. When one tank on the strain gauge sensor reached its capacity, its inflow was shut off with a solenoid valve. The liquid was then directed to the lower tank using another solenoid valve. Each measuring cylinder, with a 1000 mL capacity, was removed from the strain gauge sensors once full (to prevent overfilling). The collected water was then returned to the lower tank at the station. Water weight $w$ (N) changes in the tanks on the 15 strain gauge sensors were recorded at a constant time ($t = 1$ s). After the measurement, the water outflow rate ($W$ (N·s$^{-1}$)) was determined as the ratio of the water weight increase ($w$) to the time ($t$) taken for this increase (1). The time interval for the differential quotient was set at 60 s. After processing all the data, the 10 highest consecutive $W_{max}$ values from each monitor on the individual strain gauge sensors of leachate from the container were chosen as the 1 h percolation monitoring results. During the measurements, 45 cells were monitored in each of the four containers, and the outflow from three neighboring cells was directed to 15 individual sensors via a quadruple. A total of 600 water leachate measurement data were collected in the experiment (10 values of maximum leachate $W_{max}$ × 15 sensors × 4 containers).

$$W = \frac{(w_e - w_s)}{(t_e - t_s)} \tag{1}$$

where

$W$—increase in liquid weight over time (N·s$^{-1}$),
$w_s$—initial weight of water in the tank for time $ts$ (N),
$w_e$—final weight of water in the container for time $te$ (N),
$t_s$—start time of water mass measurement in the tank (s),
$t_e$—end time of water mass measurement in the tank (s).

The calculated $W_{max}$ was then converted into the volume of outflowing water $Q_{max}$ (m$^3$·s$^{-1}$) taking into account the liquid temperature, which was 21 °C on the day of measurement.

Then, the outflow velocity $v$ (m·s$^{-1}$) was calculated using Darcy's law (2), taking as the flow area the average inflow and outflow surfaces of liquid from a single cell of container $A$ (cm$^2$), equal to 10.4 cm$^2$ for V150 and 13.1 cm$^2$ for V300 (Table 1) [4].

$$v = \frac{Q}{A} = Ks \cdot \frac{\Delta H}{L} \tag{2}$$

where

$v$—flow rate (m·s$^{-1}$),
$Q$—outflow (m$^3$·s$^{-1}$),
$A$—total flow area perpendicular to the flow (m$^2$),
$Ks$—permeability coefficient (m·s$^{-1}$),
$L$—sample length (m),
$L'$—the level of water above container (in research $L' = 0.005$ m) determined by the outlet from frame (m),
$\Delta H$—hydraulic height (m), defined as the ratio of the height of the sample $L$ to the sum of $L$ and $L'$, (-).

For the obtained outflow velocity $v$ and the actual $ABD$ and dry bulk densities $BD$, a one-way analysis of variance was conducted to assess differences based on the container type and its repetition. Subsequently, a Pearson's linear correlation analysis ($r$) was carried out between the outflow velocity $v$ and the actual $ABD$ and dry bulk densities $DBD$ of the substrate. It was assumed that the containers in which the water drainage rate was measured had the same bulk density as those in which it was determined. For significant correlation coefficients, the relationship's strength between the two variables was assessed using the scale proposed by Guilford [20]. All statistical analyses were conducted using Statistica 12 [21].

### 3. Results

The actual and dry bulk density ($ABD$; $DBD$) was determined for selected cells in eight V150 containers (six cells each) and eight V300 containers (five cells each) (Table 2). These data reveal differences in $ABD$ and $DBD$ values between the V150 and V300 containers. Specifically, the $ABD$ in the V150 was 29.7% higher than in the V300, and 30.1% for $DBD$. The analysis of variance confirmed these differences and indicated they were associated with the container type. There was no significant effect from the repeated measurements in subsequent containers for V150 or the cell's location within the container for V150 and V300 (Table 3). This proves that the container-filling process is consistent and accurate, both in terms of space within each container and the repeatability of the process across the V150 and with some differences for V300 containers.

**Table 2.** Bulk density of the substrate filling the containers.

| Value | Container Type | |
|---|---|---|
| | **V150** | **V300** |
| Number of containers $ncABD$ and $ncDBD$ (pcs) | 8 | 8 |
| Total number of measurements in a single container $nmscABD$ and $nmscDBD$ (pcs) | 6 | 5 |
| Total number of measurements in containers $nmABD$ and $nmDBD$ (pcs) | 48 | 40 |
| Average value $ABD$ (g·cm$^{-3}$) | 0.418 | 0.322 |
| St. Dev. $ABD$ (g·cm$^{-3}$) | 0.020 | 0.021 |
| Variation coefficient $ABD$ (%) | 4.71 | 6.50 |
| Average value $DBD$ (g·cm$^{-3}$) | 0.103 | 0.079 |
| St. Dev. $DBD$ (g·cm$^{-3}$) | 0.007 | 0.006 |
| Variation coefficient $DBD$ (%) | 6.911 | 7.797 |

**Table 3.** Effect of container type, cell location, and container repetition on bulk density (average ± st. error).

| | Factor | | | | |
| --- | --- | --- | --- | --- | --- |
| | **Container Type** | **Repeating of Container** | | **Location of the Cell** | |
| | **V150 vs. V300** | **V150** | **V300** | **V150** | **V300** |
| Actual bulk density *ABD* | 0.418 ± 0.003 [b] <br> 0.322 ± 0.003 [a] | 0.400 ± 0.011 <br> 0.419 ± 0.006 <br> 0.424 ± 0.009 <br> 0.428 ± 0.006 <br> 0.420 ± 0.006 <br> 0.416 ± 0.011 <br> 0.420 ± 0.006 <br> 0.416 ± 0.007 | 0.355 ± 0.008 <br> 0.331 ± 0.007 <br> 0.326 ± 0.008 <br> 0.325 ± 0.008 <br> 0.316 ± 0.003 <br> 0.312 ± 0.008 <br> 0.308 ± 0.003 <br> 0.304 ± 0.009 | 0.413 ± 0.009 <br> 0.412 ± 0.010 <br> 0.412 ± 0.005 <br> 0.417 ± 0.006 <br> 0.430 ± 0.003 <br> 0.423 ± 0.007 | 0.308 ± 0.006 <br> 0.327 ± 0.008 <br> 0.323 ± 0.009 <br> 0.326 ± 0.007 <br> 0.326 ± 0.006 |
| | $F = 476$ <br> $p = 0.000$ ** | $F = 1.076$ <br> $p = 0.397$ | $F = 5.059$ <br> $p = 0.060$ | $F = 1.173$ <br> $p = 0.338$ | $F = 1.108$ <br> $p = 0.36$ |
| Dry bulk density *DBD* | 0.102 ± 0.001 [b] <br> 0.078 ± 0.001 [a] | 0.097 ± 0.003 <br> 0.102 ± 0.001 <br> 0.103 ± 0.002 <br> 0.104 ± 0.001 <br> 0.102 ± 0.001 <br> 0.101 ± 0.003 <br> 0.102 ± 0.002 <br> 0.101 ± 0.002 | 0.080 ± 0.002 <br> 0.080 ± 0.002 <br> 0.078 ± 0.002 <br> 0.078 ± 0.002 <br> 0.076 ± 0.001 <br> 0.075 ± 0.002 <br> 0.074 ± 0.001 <br> 0.073 ± 0.002 | 0.100 ± 0.002 <br> 0.100 ± 0.002 <br> 0.100 ± 0.001 <br> 0.101 ± 0.001 <br> 0.105 ± 0.001 <br> 0.103 ± 0.002 | 0.074 ± 0.001 <br> 0.079 ± 0.002 <br> 0.078 ± 0.002 <br> 0.079 ± 0.002 <br> 0.079 ± 0.002 |
| | $F = 507.0$ <br> $p = 0.000$ ** | $F = 1.031$ <br> $p = 0.287$ | $F = 5.039$ <br> $p = 0.057$ | $F = 1.195$ <br> $p = 0.254$ | $F = 1.110$ <br> $p = 0.325$ |

Significant differences were marked "**" < 0.01, [ab] denote homogeneous groups.

Figure 5 illustrates the changes in the weight *w* of the liquid over time *t* (60 min) for three cells of the V150 container as measured at the prototype station. Figure 6 shows the course for the first cycle of filling the measuring container (spanning about 6 min). Accompanying this is a straight line that represents the rate of weight increase as a function of time, denoted as the difference quotient W.

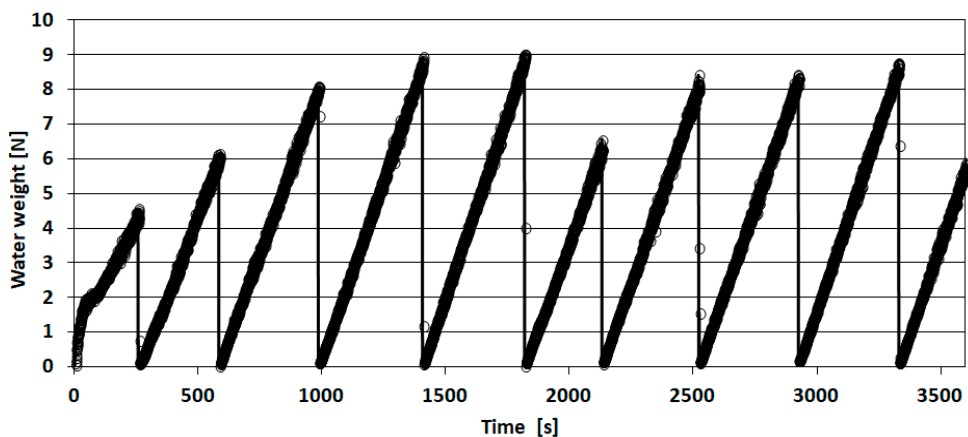

**Figure 5.** The course of changes in water weight within 1 h recorded for a single strain gauge sensor.

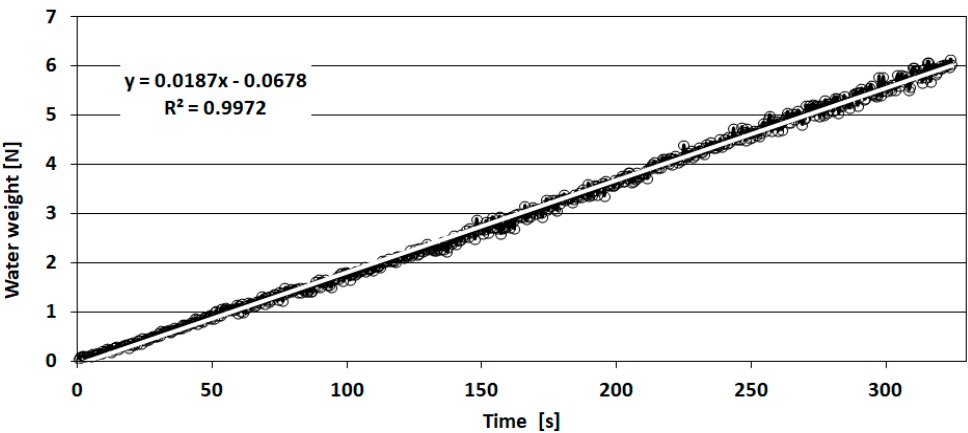

**Figure 6.** Course of water weight changes over 350 s recorded for a single strain gauge.

From an analysis of the waveforms across all 15 station sensors, water outflow velocities ($v$) and permeability coefficient ($Ks$) were determined (Table 4). Notably, the speed of water outflow ($v$) from the substrate (Table 5) differed between container types. In the V300, $v$ was 36.7% higher than in the V150. These variations were attributed to the type of container. There were no observed differences in $v$ resulting from repeated measurements in different containers or the location of the cell within the container for V150, and there were no differences for repeated measurements in different containers or the location of the cell within the container for V300.

**Table 4.** Water outflow velocities in individual containers.

| Value | Container Type | |
|---|---|---|
| | **V150** | **V300** |
| Number of containers $ncv$ (pcs) | 2 | 2 |
| Total number of measurements in a single container $nmscv$ (pcs) | 45 | 45 |
| Number of load cells (pcs) | 15 | 15 |
| Total number of measurements in containers $nmv$ (pcs) | 600 | 600 |
| Average value $v$ (cm·sek$^{-1}$) | 0.0252 | 0.0344 |
| St. Dev. $v$ (cm·sek$^{-1}$) | 0.0085 | 0.0068 |
| Variation coefficient $v$ (%) | 33.6 | 19.6 |
| Average value $Ks$ (cm·sek$^{-1}$) | 0.0244 | 0.0335 |
| St. Dev. $Ks$ (cm·sek$^{-1}$) | 0.0082 | 0.0066 |
| Variation coefficient $Ks$ (%) | 33.44 | 19.63 |

**Table 5.** Effect of container type, cell location, and container repetition on water outflow rate velocity (average ± st. error).

| | Factor | | | | |
|---|---|---|---|---|---|
| | **Container Type** | **Repeating** | | **Location of the Cell** | |
| | **V150 vs. V300** | **V150** | **V300** | **V150** | **V300** |
| | 0.025 ± 0.008 | 0.025 ± 0.001 | 0.0321 ± 0.000 [a] | 0.021 ± 0.001 | 0.037 ± 0.002 [b] |
| | 0.034 ± 0.007 | 0.026 ± 0.001 | 0.0368 ± 0.001 [b] | 0.037 ± 0.001 | 0.034 ± 0.001 [abc] |
| | | | | 0.032 ± 0.001 | 0.041 ± 0.001 [c] |
| | | | | 0.026 ± 0.002 | 0.034 ± 0.001 [ab] |
| Outflow velocities $v$ (cm·s$^{-1}$) | | | | 0.019 ± 0.001 | 0.034 ± 0.001 [abc] |
| | | | | 0.020 ± 0.001 | 0.030 ± 0.002 [a] |
| | | | | 0.023 ± 0.001 | 0.034 ± 0.002 [ab] |
| | | | | 0.027 ± 0.002 | 0.034 ± 0.002 [abc] |

**Table 5.** *Cont.*

| | Factor | | | |
|---|---|---|---|---|
| **Container Type** | **Repeating** | | **Location of the Cell** | |
| | | | 0.023 ± 0.000 | 0.036 ± 0.001 ab |
| | | | 0.030 ± 0.001 | 0.031 ± 0.001 ab |
| | | | 0.017 ± 0.001 | 0.035 ± 0.000 abc |
| | | | 0.020 ± 0.001 | 0.031 ± 0.001 c |
| | | | 0.032 ± 0.001 | 0.041 ± 0.001 ab |
| | | | 0.020 ± 0.002 | 0.032 ± 0.001 ac |
| | | | 0.029 ± 0.003 | 0.033 ± 0.003 ab |
| $F = 219.3$ $p = 0.000$ ** | $F = 2.002$ $p = 0.158$ | $F = 39.94$ $p = 0.000$ ** | $F = 12.21$ $p = 0.060$ | $F = 4.954$ $p = 0.000$ ** |

Significant differences were marked "**" < 0.01, abc denote homogeneous groups.

Assuming that the distribution of the *ABD* and *DBD* in the containers, where the water outflow velocity *v* was determined, mirrors that in the containers where *ABD* and *DBD* were ascertained, it was demonstrated that these parameters are correlated, as detailed in Table 6 and on Figure 7.

**Table 6.** Results of correlation analysis of water outflow velocity and substrate bulk density.

| | **Actual Bulk Density *ABD*** | | **Dry Bulk Density *DBD*** | |
|---|---|---|---|---|
| Outflow velocities *v* | $r = -0.524$ | $p = 0.000$ ** | $r = -0.523$ | $p = 0.000$ ** |
| Permeability coefficient *Ks* | $r = -0.531$ | $p = 0.000$ ** | $r = -0.529$ | $p = 0.000$ ** |

Significant differences were marked "**" < 0.01.

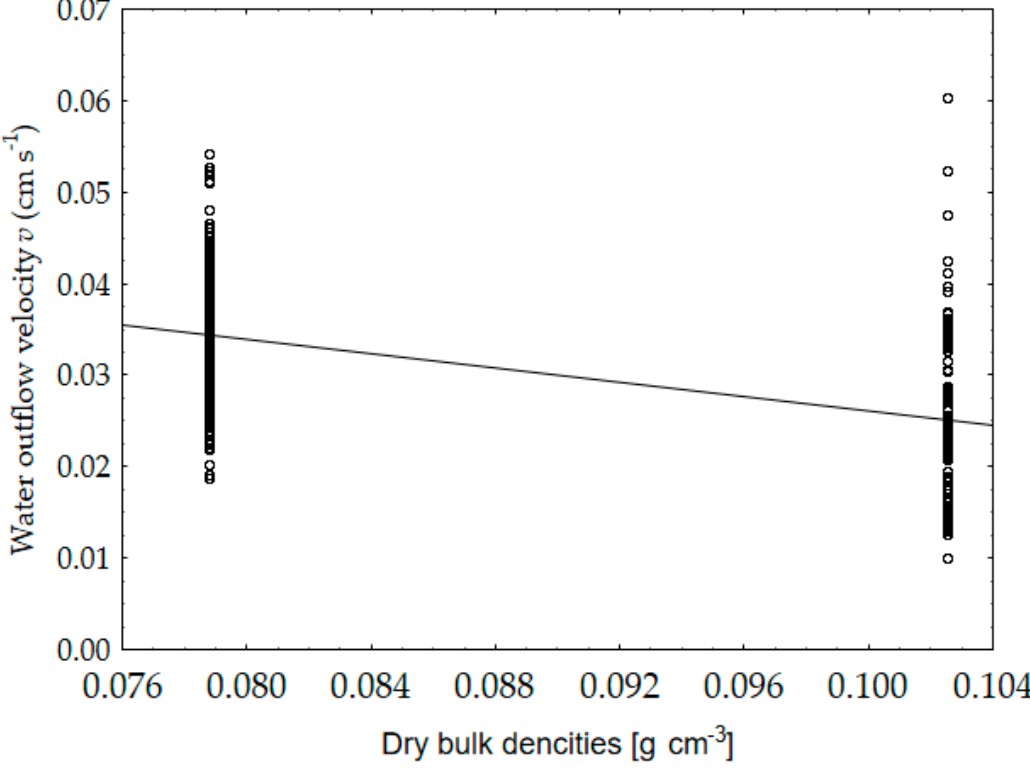

**Figure 7.** Dependence of the water outflow velocity *v* on the actual *ABD* and dry bulk density *DBD* of the substrate.

## 4. Discussion

One of the important physical parameters of soil, especially relevant when considering the movement of water through soil layers, is the flow velocity $v$. This parameter describes the ratio of the volume of flowing water $Q$ to the surface $A$ perpendicular to the flow direction. According to Darcy's law (2), $v$ is influenced by the material's permeability coefficient $Ks$, flow path $L$, and hydrostatic hydraulic height $\Delta H$. The outflow speed $v$ is vital for agricultural and forest soils due to its association with the drainage of excess rainwater into deeper soil layers and eventually to groundwater. When permeability is low, either from natural impervious layers, compacted soils, or human-made factors, the outflow speed diminishes, potentially causing ponding or surface runoff. Such situations occur with intensive use in agricultural areas [22] and forest areas [23–26]. In container production, where the container and a minimal amount of substrate are elevated from the ground, water outflows freely. Water inflow arises either from natural rainfall or artificially via ramp systems for irrigation and fertilization. In this case, the outflow speed is related to the type of substrate and the shape (surface, height) of the container cell in which the substrate is located. As determined using the proposed prototype stand, the water outflow velocity $v$ from the examined V150 and V300 containers was significantly influenced by the substrate's *ABD* ($r = -0.524$) and *DBD* ($r = -0.523$). This correlation arose from notable bulk density differences: the larger V300 cell exhibited a lower density (*ABD* = 0.322 g·cm$^3$, *DBD* = 0.079 g·cm$^3$) compared to the denser V150 cell (*ABD* = 0.418 g·cm$^3$, *BD* = 0.103 g·cm$^3$). Consequently, $v$ was affected more by container type than repetition or cell distribution. Differences in $v$ across container types were likely due to the filling time and vibration needed for the V300's larger cell versus the V150 and the compaction susceptibility of the more considerable substrate mass on a vibrating table. Both the size of the inlet surfaces (V150 $-16.64$ cm$^2$; V300 $-21.2$ cm$^2$) and the cell heights (V150 $-15$ cm; V300 $-18$ cm) might have played significant roles (Table 1). Due to the fact that both container types had consistent filling times and vibration levels (constant line efficiency of 400 containers per hour and maximum vibration acceleration of 12 G), the substrate in V300 cells did not achieve the same bulk density as in V150. There was also variation in the measured parameters in individual containers (*ABD*; *DBD*; $v$) and within the container ($v$) for V300. Therefore, line operating speeds and/or vibrating table intensities should be selected for specific container types. The demonstrated correlation between bulk density and liquid outflow speed may be of key importance when considering the relationship between substrate density and the growth of plants in containers. As the literature indicates, both too-high and too-low density values in a container cell may affect the production effect in the form of differentiated seedlings. This is particularly visible in the high variability of parameters such as shoot height, root collar diameter, root system architecture, and the degree of root overgrowth [27,28]. Individual species have different preferences as to substrate density, which may be caused by the availability of water and fertilizers that the plant can absorb before they flow out of the container cell. For example, research carried out for Scots pine (*Pinus sylvestris* L.), a major species cultivated in Poland (comprising 58.6% of its forested area) [29], indicates its heightened sensitivity to substrate compaction. The density level of pine significantly influences its growth attributes, including height, root collar thickness, dry mass of needles, shoots, and roots, and the average length of skeletal roots (>2 mm in diameter) and fine roots [27]. Both excessively high and low densities restrict the growth of this species' seedlings. Conversely, for the common beech (*Fagus sylvatica* L.), another prominent species in Poland covering 8% of the forested area [29], Pająk et al. [28] showed that high substrate density in containers negatively affects the growth of seedlings of this species. Other studies by Pająk et al. [30,31] indicate that for both pine and beech, changing the substrate density in nursery containers influenced the content of macroelements in seedlings, and high density causes a reduction in the uptake of elements, especially a reduction in the content of macroelements in the assimilation apparatus. The reason for this may be the rapid outflow of elements and the short availability time for the plant root system. That is why the density of the

substrate in the container cell is so important. This density–runoff velocity relationship might also be crucial for seed germination. Typically, low and variable bulk densities cause uneven germination since seeds, upon intensive irrigation, shift to varying depths, naturally settling due to gravity. They subsequently access water differently, contingent on substrate water retention [11,32]. Low density may cause water to drain out quickly, which may result in low moisture around the seed in the event of germination because the loose substrate releases water quickly due to its high outflow rate, and the moisture around the seed may be low or short-lived. In turn, too much water at high density and poor outflow from the cell may favor the appearance of pathogenic factors, especially those related to the increased number of fungi, as indicated by [33]. The data underscore the significance of correct bulk density level (which can be influenced) in container production. This influences the water outflow rate from containers, consequently shaping optimal plant growth conditions. A judiciously selected bulk density can also streamline water and fertilizer usage, curtail chemical runoff from containers, and minimize groundwater contamination. With well-planned and carefully considered irrigation, fertilization and monitoring of the rhizosphere and substrate moisture can yield high-quality seedlings, ensuring optimized costs of container production and minimization of groundwater contamination. This pertains to tunnel cultivation and open-area cultivation alike, as demonstrated for black spruce (*Picea mariana* (Mill.) Britton, Sterns, and Poggenb.) [34] and for white spruce (*Picea glauca* (Moench) Voss [35] and Stowe et al. [36]). However, as shown by [37], excessive irrigation and rainfall can cause nutrient losses in container production. The proposed technical solution, beyond regulating the liquid outflow velocity $v$ in individual existing nursery containers, can guide the design of containers with varied shapes, volumes, or dimensions. This device also facilitates measurements on container plants throughout their growth phases, allowing for intermittent $v$ parameter monitoring during production without the need to damage the growing plants. This study confirmed that using the Urbinati Ypsilon line to fill different container types with consistent performance parameters and vibration intensities results in variable substrate *BD* and subsequent liquid outflow velocities. Hence, filling parameters should be selected individually for specific container types.

## 5. Conclusions

Based on tests and analysis of measurement results, the following was found:

- A prototype measurement station, designed to determine the outflow of liquid from container cells filled on the Urbinati Ypsylon automated line, successfully identified the maximum velocity of liquid outflow $v$ from a substrate saturated in multiple cells of containers ranging in volume from 0.145 to 0.275 dm$^3$. The velocities ranged from 0.0252 to 0.0344 cm·s$^{-1}$.
- The observed differences in outflow velocity $v$ were associated with containers of different types, which corresponded to various bulk density values.
- With consistent settings in line efficiency (number of containers per hour) and the vibration intensity of the vibrating table, containers with different cell volumes showed variations in bulk density. This, in turn, influenced the liquid outflow speed from the substrate in the container $v$.
- Differences in the outflow velocity $v$ within the container, as determined by measurements at the station, can be used to assess the quality of the substrate, the efficiency of the automated line, and the performance of the line operators.

## 6. Patents

P.443675: *Stanowisko do pomiaru ilości odciekającej cieczy z substratu, zwłaszcza z komórek kontenerów szkółkarskich.* P.443675: A station for measuring the amount of liquid draining from the substrate, especially from the cells of nursery containers. Patent pending on 2 February 2022. Creators: Kormanek M., Małek S., Patent Office of the Republic of Poland Warsaw. 2022, pp. 12. In Polish.

**Author Contributions:** M.K.: Conceptualization, Methodology, Software, Validation, Formal analysis, Investigation, Resources, Data Curation, Writing—Original Draft, Writing—Review and editing, Visualization. S.M.: Conceptualization, Writing—Review and editing, Funding acquisition, Supervision, Project administration. All authors have read and agreed to the published version of the manuscript.

**Funding:** This research was funded by NCBiR project 1/4.1.4/2020_Application projects, Development of innovative technology to produce substrate and fertilizers from domestic raw materials for the production of forest tree seedlings. The project is cofinanced by the European Union from the funds of the European Regional Development Fund under the Smart Growth Operational Program.

**Data Availability Statement:** Data are contained within the article.

**Acknowledgments:** Special thanks to the staff of Daleszyce Forest District and thanks to the managers of the Nursery Farm in Sukowie Papiernia.

**Conflicts of Interest:** The authors declare no conflict of interest.

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
