# Peer review of "Analysis of the Water Leakage Rate from the Cells of Nursery Containers"

_forests, doi:10.3390/f14112246_

Round 1

Reviewer 1 Report

Comments and Suggestions for Authors

This study presents a device designed to measure the flow of water through substrate plugs in two different types of containers (V150 & V300) under saturated conditions. It is based on the approach of the infiltrometer and the different characteristics and parts of the measuring device are well described. A relationship is established between water outflow rate and the bulk density of substrate in V150 and V300 containers. The article highlights (i) different factors associated to the potting machine / filling line system and cell dimensions (e.g. height, volume) which may explain the variations in bulk density between containers’ type, [observed water outflow rates being consistent with measured substrate bulk densities: higher rates at lower bulk densities]; and (ii) importance of measuring the water leakage rate from cells of nursery containers to assess the quality of the potting operations and guide nursery growers with their irrigation and fertilization. The idea of relying on the physical properties of peat substrates (hydraulic and aeration) to optimize irrigation or fertigation (amount, timing, etc.) is well established and supported by numerous studies. These physical properties do evolve over time, as highlighted in the introduction, and the authors suggest using this apparatus to possibly reassess the water outflow rate from containers during the growing season. 

However, there are a few major issues with this study, in particular the bulk density. Using the weight of wet substrate to determine bulk density is not common in soil science and in the industry, mainly because its determination is affected by moisture content in the substrate when measured. Moreover, the volume of substrate in the cell was measured by dropping the cell’s content in a graduated cylinder (by reversing the container upside-down), a procedure which will change the structure of the wet substrate and thus its volume. It is not clear why the volume of substrate was not measured directly within cells. Surprisingly, the mean bulk density value for V150 is ~0.418 g cm–3 whereas in Kormanek et al. (2023), it was 0.342 g cm–3 for V150 using the same Ypsilon line I believe, hence pointing toward significant differences in BD not only between container types but also for different potting days, conditions, etc.

In my opinion, the paper should provide a better explanation for focusing mainly on water outflow rate rather than saturated hydraulic conductivity (Ks), a physical property typically reported in the literature and commonly used for comparing different studies or substrate composition. The determination of Ks at the operational scale using the device described in this paper constitutes a nice complement to measurements of bulk density.

It would be interesting to validate that the different components (filter, solenoids, etc.) have no impact on measured flow rate (e.g. with glass beads as a substrate?). Finally, there were several inconsistencies in values reported in Tables and statements in the Discussion (see my comments in the attached PDF file).

Author Response

Dear Editor, Reviewers

We have made changes to the article in accordance with all suggestions contained in Reviews 1 and 2. All changes are marked in yellow, and information about which reviewer was changed was added.

Below are responses to Reviewers' comments.

Best regards

Mariusz Kormanek

Dear Reviewer, thank you very much for your thorough review of the publication. Below are responses to the reviewer's comments:

Reviewer 1 comments:

  1. However, there are a few major issues with this study, in particular the bulk density. Using the weight of wet substrate to determine bulk density is not common in soil science and in the industry, mainly because its determination is affected by moisture content in the substrate when measured.

Reply Authors

In the work, a parameter of dry bulk density was added, marked as DBD, and the parameter of actual bulk density was renamed on ABD

  1. Moreover, the volume of substrate in the cell was measured by dropping the cell’s content in a graduated cylinder (by reversing the container upside-down), a procedure which will change the structure of the wet substrate and thus its volume. It is not clear why the volume of substrate was not measured directly within cells.

Reply Authors

The adopted method of determining bulk density is correct. Pouring the mass of wet substrate using a special pick-up and cylinders inserted in it, the position of which corresponds to the outlet of a single cell and rotated with the container by 180 degrees, enabled quick measurement of the mass of the substrate in individual cells. Knowing this mass and the volume of the cells (from producer) and knowing that the substrate completely filled the entire cells, the bulk density was calculated. Collecting the substrate in a different way, which we also did using e.g. a spoon, is difficult and time-consuming, and does not allow for quick measurement in many cells. This method allows you to perform the measurement quickly - all you need to do is measure the masses of the cylinders with the substrate and know the mass of the cylinder.

  1. Surprisingly, the mean bulk density value for V150 is ~0.418 g cm–3 whereas in Kormanek et al. (2023), it was 0.342 g cm–3 for V150 using the same Ypsilon line I believe, hence pointing toward significant differences in BD not only between container types but also for different potting days, conditions, etc.

Reply Authors

Yes, we agree with this conclusion, differences appear, which result from differences in the substrate itself, which is not perfectly the same in each big bag (e.g 1,2-1,5 m3, or bag -200 liters) (it is influenced by moisture in the bag, storage method, where in the pile on the pallet the bag was located at the bottom or on top), from mixing and moisturising in the mixer, from how the people working on the line perform their work. However, the differences in compaction due to the type of container with the same parameters during backfilling were confirmed as in Kormanek at al 2023. In the V150 BD containers it was higher than the V 300. The main point of the work was to draw attention to the fact that using the same settings for different containers, which is commonly used in nurseries is not good and the procedure for preparing containers on the line. It should be individualized depending on the type of container and perhaps the species that will be grown, as indicated by the measurement results presented in the PajÄ…k 2022a-d publications cited in the work.

  1. In my opinion, the paper should provide a better explanation for focusing mainly on water outflow rate rather than saturated hydraulic conductivity (Ks), a physical property typically reported in the literature and commonly used for comparing different studies or substrate composition. The determination of Ks at the operational scale using the device described in this paper constitutes a nice complement to measurements of bulk density.

Reply Authors

We also believe that it is more important to pay attention to the outflow velocity v, which is crucial in order to show the problem of the impact of the procedure of filling containers on the loss of water or elements from fertilizers flowing out of the cells. The Ks parameter was added in the work, but due to the different determination methodology compared to the laboratory method using permeability meters, e.g. Eijkelkamp, it is difficult to interpret. Ks is usually determined in a standardized cylinder fed with water from the bottom, which is not the case in a nursery container fed from the top and in a cell with a different shape and free drainage due to air cutting of the roots. We wanted to present a real drain in a container in which cells have a specific shape and in which there is a substrate thickened to parameters similar to those in real production conditions.

  1. It would be interesting to validate that the different components (filter, solenoids, etc.) have no impact on measured flow rate (e.g. with glass beads as a substrate?).

Reply Authors

Indeed, it would be possible to check whether the filter or solenoid does not affect the flow with the method proposed by the reviewer. However, we believe that this influence is minimal, because the outflow is naturally gravity-fed without being blocked by solenoid valves, which are only used during measurement to change the direction of flow. The filter drains after filling the filter tank with water and reaching the level of the drain connector. The use of a settling filter was caused by the desire to determine what particles are washed out from the substrate. Please also note that this is a prototype that may be further modified to improve its functionality and quality of operation. Filter descriptions have been supplemented in the text.

  1. Finally, there were several inconsistencies in values reported in Tables and statements in the Discussion (see my comments in the attached PDF file).

Reply Authors

All these comments were addressed in the publication in response to questions

  1. It is not clear why many data points have a similar BD value (i.e., 0.418 and 0.322 g cm-3; means shown in Table 2) while others are spanning a larger range of BD values. Moreover, based on Table 2, there were 48 (i.e. 8 containers x 6 cells) and 40 measurements for V150 and V300 containers, respectively. However, the number of data points shown here exceeds these values. Why?

Reply Authors

Average data from dry bulk dencities DBD were taken as the x-axis data, and data from 15 sensors with 10 samples of the difference quotient from each measurement lasting 1 h were taken as the data for the water outflow rate, as described in the paper. It was also corrected the Fig. 5.

Dear Reviewer, thank you very much for your thorough review of the publication. Below are responses to the reviewer's comments:

Reviewer 2 comments:

  1. In the MATERIALS AND METHODS, author mentioned that “The experiment utilized V150 and V300 Styrofoam 95 containers manufactured by Marbet ”, Why did the author choose these two models of containers? Are these two types of containers commonly used locally?

Authors' responses

This information has been supplemented in the text of the publication:

.... These type of containers are commonly used in Poland for the production of coniferous species, e.g. pine, spruce (V150) and deciduous species, e.g. beech and oak (V300).

  1. In the Materials and Methods, the containers were filled with peat–perlite substrate (95/5 by volume), why use this matrix ratio and what is the basis for selection?

Authors' responses

This information has been supplemented in the text of the publication:

.....This substrate is produced in Poland based on imported peat, and the percentage of perlite added is determined individually for each batch of peat delivery, based on the analysis of air and water capacity.

  1. In the Material Method, “a one-way analysis of variance was conducted to assess differences”, since the analysis has been conducted, it is recommended to add standard error and significance analysis after each group of measured data .

Authors' responses

Corrections were made, everything was recalculated, tables were changed again, information was added according to the reviewer's suggestion.

  1. It is recommended to further beautify Figures 5, 6, and 7, with the scale lines facing inward or outward, the thickness of the lines, and the presence or absence of grid lines that need to be unified.

Authors' responses

Figures were corrected as suggested

  1. Please carefully review the entire text and use the correct writing method. Latin names need to be italicized, such as Scots pine (Pinus sylvestris L.), beech (Fagus sylvatica L.), etc.

Authors' responses

This error has been corrected. It is not the authors' fault, it appears when sending the file to the server, in some places italic disappears and word combinations appear. We don't know what the reason is for this.

  1. The conclusion section is not concise enough and needs to be rewritten to summarize the core results of the experiment.

Authors' responses

Conclusions were shortened as suggested

  1. Please check all references and follow the journal's requirements for layout.

Authors' responses

References were checked and appropriate corrections were made

Reviewer 2 Report

Comments and Suggestions for Authors

In the manuscript “Analysis of the water leakage rate from the cells of nursery containers” by Kormanek et al., author have investigated the water outflow velocity of two Styrofoam container types (V150-650/312/150 mm; 74 cells; 0.145 cm3 cell volume and V300-20 650/312/180 mm; 53 cells; 0.275 cm3 cell volume) after filling with with a peat and perlite substrate (95/5%), bulk density of the substrate, effect of container type, cell location, and container repetition on bulk density, water weight within 1 hour, water weight changes over 60 s, water outflow velocities in individual containers and dependence of the water outflow velocity ν on the current bulk density BD of the substrate were analyzed, the research results have certain significance for container production, effective irrigation and optimize irrigation. However, there are still some issues that need to be explained reasonably, as follows:

1. In the MATERIALS AND METHODS, author mentioned that “The experiment utilized V150 and V300 Styrofoam 95 containers manufactured by Marbet ”, Why did the author choose these two models of containers? Are these two types of containers commonly used locally?

2. In the Materials and Methods, the containers were filled with peatperlite substrate (95/5 by volume), why use this matrix ratio and what is the basis for selection?

3. In the Material Method, a one-way analysis of variance was conducted to assess differences, since the analysis has been conducted, it is recommended to add standard error and significance analysis after each group of measured data .

4. It is recommended to further beautify Figures 5, 6, and 7, with the scale lines facing inward or outward, the thickness of the lines, and the presence or absence of grid lines that need to be unified.

5. Please carefully review the entire text and use the correct writing method. Latin names need to be italicized, such as Scots pine (Pinus sylvestris L.), beech (Fagus sylvatica L.), etc.

6. The conclusion section is not concise enough and needs to be rewritten to summarize the core results of the experiment.

7. Please check all references and follow the journal's requirements for layout.

Comments on the Quality of English Language

The writing of the paper is relatively smooth and the word order is smooth, but there are still some areas that need further optimization.

Author Response

(The authors gave the same response as above.)

Round 2

Reviewer 1 Report

Comments and Suggestions for Authors

The authors addressed all the points that I raised. I am satisfied with the changes made to the manuscript. 

Replace "sek–1" in Tables by "s–1"

Reviewer 2 Report

Comments and Suggestions for Authors

In the manuscript “Analysis of the water leakage rate from the cells of nursery containers” by Kormanek et al., author have investigated the water outflow velocity of two Styrofoam container types (V150-650/312/150 mm; 74 cells; 0.145 cm3 cell volume and V300-20 650/312/180 mm; 53 cells; 0.275 cm3 cell volume) after filling with with a peat and perlite substrate (95/5%), bulk density of the substrate, effect of container type, cell location, and container repetition on bulk density, water weight within 1 hour, water weight changes over 60 s, water outflow velocities in individual containers and dependence of the water outflow velocity ν on the current bulk density BD of the substrate were analyzed, the research results have certain significance for container production, effective irrigation and optimize irrigation. Based on the revised manuscript submitted by the author, it can be seen that the author has made detailed revisions to the article and provided reasonable explanations for any shortcomings, which can be accepted and published.